

# *G6PD* and *HBB* polymorphisms in the Senegalese population: prevalence, correlation with clinical malaria

Fatou Thiam[1,*], Gora Diop[2,3,*], Cedric Coulonges[4], Céline Derbois[5], Babacar Mbengue[6], Alassane Thiam[3], Cheikh Momar Nguer[1], Jean Francois Zagury[4], Jean-Francois Deleuze[5] and Alioune Dieye[6]

[1] Groupe de Recherche Biotechnologie Appliquée et Bioprocédés Environnementaux (GRBA-BE), Laboratoire Eau, Energie, Environnement et Procédés Industriels (LE3PI), Département de Génie Chimique et Biologie Appliquée, Ecole Supérieure Polytechnique, Université Cheikh Anta DIOP de Dakar, Dakar Fann, Dakar, Sénégal

[2] Unité Postulante de Biologie Génétique, Génomique et Bio-informatique (G2B), Département de Biologie Animale, Faculté des Sciences et Techniques, Université Cheikh Anta DIOP, Avenue Cheikh Anta DIOP, Dakar Fann, Dakar, Sénégal

[3] Pole d'Immunophysiopathologie & Maladies Infectieuses (IMI), Institut Pasteur de Dakar, Dakar, Sénégal

[4] Equipe GBA «Génomique, Bioinformatique & Applications », Conservatoire National des Arts et Métiers, Paris, France

[5] CEA, Centre National de Recherche en Génomique Humaine, Université Paris-Saclay, Evry, France

[6] Service d'Immunologie, Faculté de Médecine, de Pharmacie et d'Odontostomatologie, Université Cheikh Anta DIOP, Dakar, Sénégal

* These authors contributed equally to this work.

Corresponding author
Fatou Thiam,
fatou54.thiam@ucad.edu.sn,
fatou.thiam@esp.sn

## ABSTRACT

**Background**. Host genetic factors contribute to the variability of malaria phenotypes and can allow a better understanding of mechanisms involved in susceptibility and/or resistance to *Plasmodium falciparum* infection outcomes. Several genetic polymorphisms were reported to be prevalent among populations living in tropical malaria-endemic regions and induce protection against malaria. The present study aims to investigate the prevalence of *HBB (chr11)* and *G6PD (chrX)* deficiencies polymorphisms among Senegalese populations and their associations with the risk for severe *Plasmodium falciparum* malaria occurrence.

**Methods**. We performed a retrospective study with 437 samples, 323 patients recruited in hospitals located in three different endemic areas where malaria episodes were confirmed and 114 free malaria controls. The patients enrolled were classified into two groups: severe malaria (SM) (153 patients) and uncomplicated malaria (UM) (170 patients). PCR and DNA sequencing assessed host genetic polymorphisms in *HBB* and *G6PD*. Using a multivariate regression and additive model, estimates of the impact of human *HBB* and *G6PD* polymorphisms on malaria incidence were performed.

**Results**. Six frequent SNPs with minor allele frequencies (MAF) > 3% were detected in the *HBB* gene (rs7946748, rs7480526, rs10768683, rs35209591, HbS (rs334) and rs713040) and two in the G6PD gene (rs762515 and rs1050828 (*G6PD*-202 G > A). Analysis of selected HbS polymorphism showed significant association with protective effect against severe malaria with a significant *p-value* = 0.033 (OR 0.38, 95% CI [0.16–0.91]) for SM *vs.* UM comparison. Surprisingly, our study did not identify the protective effect of variant HbC polymorphism against severe malaria. Finally, we found some of

the polymorphisms, like HbS (rs334), are associated with age and biological parameters like eosinophils, basophils, lymphocytes etc.

**Conclusion.** Our data report *HBB* and *G6PD* polymorphisms in the Senegalese population and their correlation with severe/mild malaria and outcome. The *G6PD* and *HBB* deficiencies are widespread in West Africa endemic malaria regions such as The Gambia, Mali, and Burkina Faso. The study shows the critical role of genetic factors in malaria outcomes. Indeed, genetic markers could be good tools for malaria endemicity prognosis.

## BACKGROUND

Malaria is caused by *Plasmodium* species infection and affects hundreds of millions of people per year. In 2020, an estimated 241 million malaria cases occurred in 85 malaria-endemic countries, increasing from 227 million in 2019, with most of this increase coming from countries in the African Region (95% of cases). Malaria disease remains a significant cause of death and is still the 4th leading cause of death from infectious diseases worldwide (*World Health Organization, 2021*; *Papaioannou, Utzinger & Vounatsou, 2019*; *Moxon et al., 2020*). The symptoms caused by *Plasmodium falciparum* infection are highly variable, ranging from asymptomatic (AM) and uncomplicated (UM) to severe clinical forms (SM) (*Conway, 2007*; *Miller et al., 2002*; *Niang et al., 2016*). Reasons for these different progressions from asymptomatic, uncomplicated to severe forms or fatal outcomes are not fully understood (*Milner, 2018*). Understanding these mechanisms is necessary for global malaria eradication (*Mackinnon et al., 2005*; *Weatherall & Clegg, 2002*; *Griffiths et al., 2005*).

Host genetic factors contribute to the variability of malaria phenotypes (*Moxon et al., 2020*) and may provide insight into the mechanisms involved in susceptibility and/or resistance to *Plasmodium* species infection outcome (*Miller et al., 2002*). Several genetic polymorphisms are prevalent among populations living in tropical malaria-endemic regions and induce protection against malaria (*Miller et al., 2002*; *Niang et al., 2016*). Glucose-6-phosphate dehydrogenase (*G6PD*) and β-chain of haemoglobin (*HBB*) deficiencies are more prevalent in malaria-endemic countries and are two of the essential loci conferring resistance to severe malaria in humans (*Mackinnon et al., 2005*; *Grignard et al., 2019*).

Mutations in the *HBB* gene, located on Chromosome 11 (*Onda et al., 2005*), are responsible for several serious hemoglobinopathies, such as sickle cell anaemia and β-thalassemia (*Lopera-Mesa et al., 2015*; *Modiano et al., 2001*; *Taylor, Parobek & Fairhurst, 2012*; *Williams, 2005a*; *Williams, 2015*; *Williams, 2005b*). To date, there are more than 900 variants in *HBB* (*Aldakeel et al., 2020*), and in malaria-endemic areas, several HBB variants are known to have a protective effect against *P. falciparum* malaria (*Grant et al., 2015*). HbAS variant has been associated with 50% and 80% reduced risks of developing uncomplicated and severe malaria (*Williams, 2005b*), HbAC and HbC variants have

been associated with 30% and 93% reduced risks of developing mild and severe malaria, respectively (*Modiano et al., 2001*; *Taylor, Parobek & Fairhurst, 2012*; *Agarwal et al., 2000*; *Travassos et al., 2015*). HbC is prevalent only in West Africa (*Tetard et al., 2017*; *Piel et al., 2013a*), and allele frequencies above 15% have been described in West African populations (*Piel et al., 2013b*). The sickle hemoglobin HbS is encountered at frequencies of up to 18% across sub-Saharan Africa, the Middle East and South Asia. HbSS homozygous subjects develop sickle-cell anaemia disease associated with low life expectancy. Generally, subjects succumb in early life; unlike the HbCC homozygosity, the HbAS and HbAC heterozygous traits are clinically benign (*Piel et al., 2013a*).

*G6PD* is encoded by a 16.2 kb gene found on the X chromosome (*Gomez-Manzo et al., 2016*). Approximately 160 genetic variants causing clinical deficiency of *G6PD* have been characterised (*Mason, Bautista & Gilsanz, 2007*). *G6PD* deficiency affects over 400 million people living in tropical and subtropical countries, 15–30% of whom are founded in sub-Saharan Africa (*Lwanira et al., 2017*; *Carter et al., 2011*; *Clarke et al., 2017*). A high diversity of *G6PD* variants has been reported, including the most common form *G6PD* B (wild type enzyme), characterised by the 202G-376A allele. In sub-Saharan Africa, up to 40% of the population carries the *G6PD* A+ form (non-deficient type) characterised by 202G-376G allele with 85% enzyme activity (*Battistuzzi et al., 1977*). With the presence of an added mutation at the 202 nucleotides (G > A), the double mutant (202A-376G) is named *G6PD* A- with only 12% enzyme activity compared to the B allele (*Beutler, 1989*). The 202A- allele has been associated with protection against severe malaria in African populations in Mali, the Gambia, and Uganda (*Guindo et al., 2007*; *Ruwende et al., 1995*; *Walakira et al., 2017*). However, this association was less significant than sickle haemoglobin and α-thalassemia (*Verra, Mangano & Modiano, 2009*).

The high prevalence of malaria protective polymorphisms is clearly associated with endemicity, but the prevalence varies among populations living in endemic areas. Local malaria risk can explain some of these differences, as suggested by the decreased prevalence of protective polymorphisms with increasing altitude in endemic countries (*Hedrick, 2011*). The prevalence of some protective polymorphisms has also been shown to vary between ethnic groups (*Hedrick, 2011*).

In Senegal, the polymorphisms of *HBB* and *G6PD* have already been investigated. Main studies focus on a small number of participants and/or ethnic groups. Since 1960, some data have been obtained concerning the prevalence of HbS and *G6PD* polymorphisms in Bedik and Niokholonko populations in the Southeastern part (*Mauran-Sendrail et al., 1975*). The study of *G6PD* from the Niokholonko ethnic group reported the frequencies of two variants of *G6PD*: 0.18 for *G6PD* A+ and 0.07 for *G6PD* A- (*Vergnes, Gherardi & Bouloux, 1975*). It also appears that in the Sereer ethnic group, from the Niakhar area located 150 km southeast of Dakar, Senegal, the allele *G6PD*-376G/202A is not the common variant but rather the 376G/968C variant (*De Araujo et al., 2006*). A study performed in 2005 in the Dakar region focused on 300 individuals described the prevalence and morbidity of *G6PD* Deficiency in the Sickle Cell Disease homozygote group (*Diop et al., 2005*). In neighbouring countries such as the Gambia, it was reported that the minor allele frequency

of *G6PD*-202A in healthy participants was 0.03, considerably lower than those reported in eastern African countries such as Kenya (0.18) and Malawi (0.19) (*Clarke et al., 2017*).

However, until now, there has been a lack of data about the frequencies of the general Senegalese population, from which full studies have not yet been performed. It is essential to perform additional screening of *HBB* and *G6PD* loci to identify potential variants associated with susceptibility to severe malaria and corresponding hematologic parameters.

Senegal is a West African country where malaria risk varies from one region to another. Our present study has two aims: (i) first, to investigate the prevalence of key malaria-protective polymorphisms in *G6PD* and *HBB* genes in the Senegalese population, (ii) secondly, we will check the correlation between main polymorphisms in *HBB* and *G6PD* genes and severe malaria outcome. In participants from three ecological zones, we characterised polymorphisms on HBB and G6PD genes where the malaria endemicity was different. More data on *G6PD* and *HBB* polymorphisms and their incidence on malaria outcome and corresponding haematological parameters are essential for different malaria eradication strategies in endemic West African countries.

## MATERIAL AND METHODS

### Senegalese population, ethnical distribution and study sites

The cohort included black Senegalese-born individuals recruited between 2003 and 2015, whose parents and grandparents were born in Senegal, a malaria-endemic country in the Sahelian zone of West Africa.

The Senegalese population (SP) recorded in 2013 is 13.508.715 inhabitants, including near equality between men 49.9% and women 50.1% (*Agence Nationale de la Statistique et de la Démographie (ANSD), 2014*). SP is unevenly distributed in space and is concentrated in the West and Center of the Country, while the East and the North are sparsely populated. Then the highest density of population is found in cities such as Dakar, the Capital, Thies, Touba, Saint-Louis-Louga, and Kaolack.

Senegalese ethnic groups are located almost everywhere in Senegal, but the 'Wolof Ethnic group' is concentrated in the West and West-Center; the 'Serere populations' are focused on Peanut-Basin, specifically in Kaolack and Fatick regions. The Peulh ethnic groups are scattered throughout Senegal areas but are more established in Ferlo along the river and the Kolda region. The 'Sarakholes' live in Bakel. Other ethnic groups, such as Diolas, Baïnouks, Balantes and Madingues Ethnic groups, are located in Casamance. Bedik and Niokholonko populations are located in the country's Southeastern part, such as Tambacounda and Kedougou regions.

Clinical Malaria participants were enrolled from five "site-regions" in Senegalese regional hospitals in St Louis and Louga in the North, Dakar, Tambacounda and Kolda (Fig. 1). These targeted urban cities polarise different ethnic groups, particularly in Dakar, the capital of Senegal, even though there is a difference in the ethnic distribution between North to South.

For convenience, these five regions were divided into three ecological areas according to the climatic gradient and malaria endemicity, translated from North to South. The

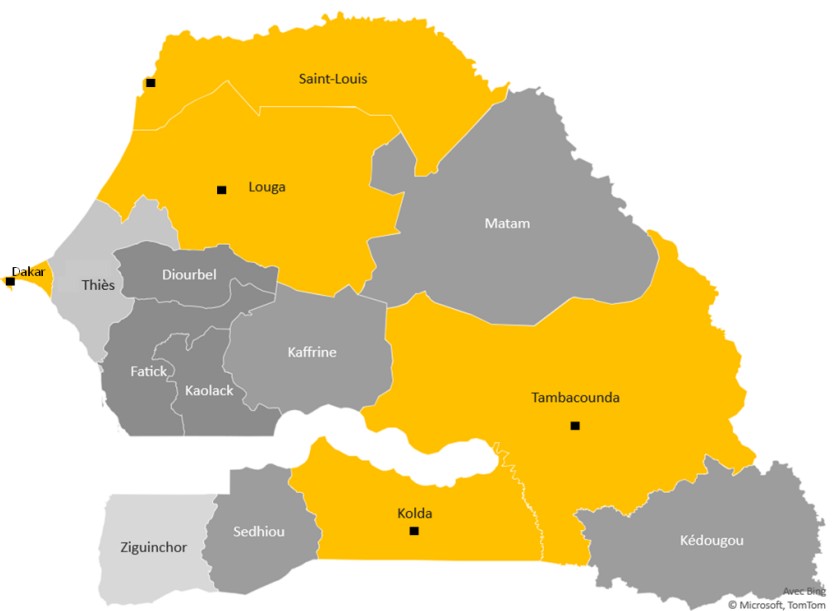

**Figure 1** **The study sites.** Study region map. Black points mark the towns surveyed in Senegal and show the study sites. The yellow color areas keep the different ecological areas of endemicity, in north the pree-limination area: St Louis-Louga regions, the capital Dakar-Diamniadio center and the south endemic region Tambacounda-Kolda.

northern part is Sahelian climates with unstable or seasonal malaria, including St Louis and Louga regions. These sites belong to the ecological area where malaria is hypoendemic and where the annual incidence in 2019 was less than five cases per 1,000 inhabitants. Some sites are under 1/1000 with very low anopheles vector density. This region is located in the malaria prelimination area context (*Thiam et al., 2021*; *Thiam et al., 2020*). The 'centre' part is Dakar, an urban centre with low malaria prevalence rates and better health coverage due to better healthcare access. The southern region includes Kolda, and Tambacounda and corresponds to the Sudano-Guinean zone, with annual rainfall exceeding 800 mm (*Sahondra Harisoa et al., 2001*). Until now, malaria transmission has been hyperendemic in southern regions, and the high transmission season in Senegal occurs mainly between July and October (*Jambou et al., 2001*; *Diouf et al., 2017*).

## Sampling, clinical and controls participants

Clinical malaria participants corresponded to participants with Plasmodium-positive QBC (Quantitative Buffy Coat). The QBC is a qualitative test, conceptualised as far back as 1974, and is used to identify the malarial parasites in the peripheral blood faster than the conventional thick smear method. The technique involves staining the centrifuged and compressed red cell layer with acridine orange, which is later examined under a fluorescence microscope (*Benito et al., 1994*; *Prashanth, 2012*).

According to the criteria defined by *Saissy, Rouvin & Koulmann (2003)* the malaria participants were assigned to two groups: uncomplicated malaria (UM) and severe malaria (SM). 'Severe malaria' phenotype is characterised by the cerebral form, an organ failure

and/or metabolic dysfunctions secondary to *P. falciparum* infection. High parasitemia, Severe anaemia and cerebral form are the three most 'severe forms' currently explored. But other complications such as hypoglycaemia, thrombocytopenia, renal insufficiency, hepatic or even pulmonary oedema may appear alone or in combination. For uncomplicated forms, we considered patients who had fever with P. falciparum parasitaemia of <25,000 parasites/μL of blood with no evidence of impaired consciousness or seizures at the time of enrollment and no other past background of mental illness, meningitis or accidental head injury were included.

To ensure homogeneity of data, inclusion criteria were: (1) Only black Senegalese individuals with *P. falciparum* infection confirmed in diagnostic; (2) persons born in Senegal; whose parents and "grandparents" were born in Senegal. Exclusion criteria were: (1) other racial or ethnic groups living in Senegal; (2) participants with clinical signs of severity or any state that may interfere with the study, such as a recent pregnancy and childbirth. The control group (CTR) corresponded to unaffected participants, free of Plasmodium infection. These participants were recruited at Pasteur institute at the biomedical laboratory according to the same criteria used for clinical participants inclusion, and live in the same environmental and endemic area".

A blood sample (10 mL) was taken from venous blood in heparin-containing vacutainer tubes for each patient. Hemogram and Several blood cell indices, such as red blood cells (RBCs) count, Hematocrit, MCV (mean corpuscular volume) and MCHC (mean corpuscular volume concentration), were taken in the different groups and evaluated according to references normal values suggested elsewhere (*Lopera-Mesa et al., 2015*; *Modiano et al., 2001*). In SM and UM participants, parasitaemia h been estimated by first counting the number of parasites per 200 white blood cells in a thick blood film and then calculating the parasite count/μL from the total white blood cell count μL (*Ngom et al., 2007*).

## DNA extraction, PCR and sequencing

As previously described by *Diop et al. (2018)*, genomic DNA was extracted from peripheral blood using a standard Qiagen Kit following the manufacturer's recommendations (QIAmp kit Cat. No 51306).

Oligonucleotide primers were designed to amplify interest regions in *HBB* and *G6PD* by PCR. At all, PCR reactions were performed using a Gotaq®Green Master Mix (Promega, Germany) in a total volume of 25 μl containing 25 ng of genomic DNA (5 ng/μl) and 2.5 μl of each primer (10 μM). PCR conditions were initial denaturation at 95 °C for 5 min, 35 cycles at 95 °C for 30 s, 62 °C for 45 s, and 72 °C for 1 min, with a final extension at 72 °C for 10 min.

PCR amplification for the Hemoglobin *HBB* gene was performed using a reference sequence located at chromosome 11 (GRCh38:11:5226434-5227234). The sequence region was downloaded from the BLAST tools, and the primers positions of HbS—rs334 ([A/T]), HbC—rs33930165 ([C/T]) polymorphisms were designed. The reference sequence downloaded is orientated forward concerning the reference genome. The primers sequences

designed for *HBB* amplicons were reversed concerning the reference, such as forward (5′-ACTCCTAAGCCAGTGCCAGA-3′) and reverse (5′-CGATCCTGAGACTTCCACAC-3′) primers. An amplicon of 801 bases was obtained, targeting mutations of deficiencies, including HbS (rs334) and HbC (rs33930165) and surrounding polymorphisms.

For the *G6PD* gene, the location of the amplicon was (GRCh38: X:154535674-154536374). PCR amplification was performed with forward (5′-GTCTTCTGGGTCAGGGAT-3′) and reverse (5′-GGAGAAAGCTCTCTCTCC-3′) primers, amplifying 701 bases long and targeting the *G6PD-202* and their surrounding SNPs. Then the designed primers are reverse with respect to the human reference sequence (GRCh38: X:154535674-154536374) (http://www.ensembl.org/index.html). Then the amplicon left out SNP previously identified in the Senegalese population, such as *G6PD* +376, *G6PD* +376, *G6PD* +542, *G6PD* +680 and *G6PD* +968.

The PCR products were examined for specificity *via* 2% agarose gel electrophoresis, and the amplicons were purified using BioGel P100 gels (Bio-Rad). Sequencing reactions (2 µl of PCR product) were performed using the Dye Terminator v3.1 method in an ABI PRISMs 3730 DNA Analyzer (Applied Biosystems, Foster City, CA, USA). Sequencing conditions were: 96 °C for 5 min, 25 cycles of 96 °C for 10 s; 60 °C for 4 min and 15 °C forever, and PCR products were purified with Sephadex G50 superfine columns (GE Healthcare). Alignment and SNP discovery were performed using NG_009015.2 and NG_059281. as reference sequences for *G6PD* and *HBB,* respectively. Analysis was performed with Genalys version 2.0 software (*Takahashi et al., 2003*).

## Statistical analysis

Statistical association tests Epi Info software (version 7.0.8.1) were performed to evaluate the association between malaria clinical status and *G6PD* and *HBB* polymorphisms. At first Allelic frequencies and Hardy-Weinberg equilibrium were calculated, as described (*Rodriguez, Gaunt & Day, 2009*). The differences in allelic frequencies between the three groups (SM, UM, CTR) were determined using the logistic regression analysis method, as reported previously (*Barrett et al., 2005*; *Tregouet & Garelle, 2007*).

Associations between SNPs and different parameters such as age, biological and hematological parameters were performed using the Mann–Whitney test, and then associations with *p* values < 0.05 were considered statistically significant.

Different approaches were explored for association analysis with *G6PD* loci polymorphism to assess hemizygosity. First, a random inactivation was coded accordingly to genotype as [0, 2] for males and [0, 1, 2] for females, accounting for the random inactivation and assuming an activation of one of the two X chromosomes. Second, an 'SNP random inactivation escape' was taken into account, and a test was performed in combined samples, given that a proportion of SNPs can escape from random inactivation used at first. Then gender was treated as a covariate, and the association tests were adjusted in the discovery sample. Third, the association test for males and females was performed separately to investigate sex-specific effects underlying X chromosome loci. An association test accounting for gender × SNP interaction was also performed. Meta-analyses were then conducted using the R package (version 1.9–8) for combined discovery analyses. For

this model approach separating males and females would cause a lost statistical power according to the small size of our sampling. Finally, the heterogeneity among these datasets from the different models was evaluated by tests such as I2 and Q test > 50% or $P < 0.05$ were considered heterogeneous, and the random-effect model was applied; otherwise, the fixed-effect model would be used. Linkage disequilibrium (LD) was computed for each pair of polymorphisms within the *HBB* and *G6PD* genes using Haploview software (*Barrett et al., 2005*). The Lewontin's D′ coefficient correlates to the level of recombination, which helps find a haploblock. The LD was calculated as a summary and according to all ethnic group populations. Haplotype estimates were obtained using Shapeit4 (*Delaneau et al., 2019*). Nominal *p*-values were corrected under the cofounders association effects HbS and HbC polymorphisms.

## Ethics

The study's objectives have been explained clearly using local dialect before performing inclusion of patients in hospitals centres. The protocol has been reviewed according to the rules issued by the National Committee for Ethics for Health Research (CNERS) of Senegal and according to the procedures established by the Cheikh Anta Diop University of Dakar (UCAD) for the ethical approval of any research involving human participants. Written informed consent was obtained from adult participants and parents or legal representatives of children. In addition, based on the information provided, UCAD's Committee on Research and Ethics (CER) considers that the research proposed respects the appropriate ethical standard and, as a result, approves its execution under ''Protocole 0344/2018/CER-UCAD''. All patients enrolled in the cohort/or legal representative gave signed and informal written consent to provide a blood sample for further studies.

# RESULTS

## Study sites and populations

We performed a retrospective study on 437 participants recruited between 2003 and 2015 in three ecological areas distinguished by their malaria endemicity (Fig. 1, Table 1). Of those, 27 (6%) were from St Louis-Louga, 258 (59%) from Dakar, and 147 (33%) from Tambacounda-Kolda. In total, 153 participants had fulfilled WHO criteria for SM (*Bei et al., 2017*), 170 for UM, and 117 for CTR. Population CTR controls were recruited in the same ecological and endemic area as clinical cases, are free of 'Plasmodium' infection and represent 26.7% of the study population. The median age was 25.8 years in individuals with SM and 18.9 years in individuals with UM. Eighteen percent (18%) were children under five years of age, 15% were 5-15 years old, and 55% were above 15 years old. In our cohort, no mortality was observed in UM and CTR groups. However, 13.7% of mortality was observed in severe malaria patients, with 1.5% in children 15 years and under. 207 cases (47.36%) were male in our population, and 194 (44.39%) were female.

## Haemotological and parasitaemia parameters of the study population

The average values of selected haematological parameters were determined for three SM, UM and CTR groups and the association was calculated using ANOVA statistical

**Table 1 General Characteristics of the study participants.** The number of patients in each group SM, UM and CTR, is shown. Age is given with median values (the number of subjects and corresponding percentages is given in parenthesis). ND the number for which, data are undetermined and/or unknown.

| | | Severe Malaria (SM) (N = 153) | Uncomplicated Malaria (UM) (N = 170) | Control (CTR) (N = 114) | Total (N = 437) |
|---|---|---|---|---|---|
| Age (years) | <5, n (%) | 29 (18.9) | 45 (26.4) | 6 (5.2) | 80 (18.3) |
| | [5–15], n (%) | 29 (18.9) | 28 (16.4) | 20 (17.5) | 77 (17.6) |
| | [15–25], n (%) | 37 (24.2) | 15 (8.8) | 6 (5.2) | 58 (13.3) |
| | >25, n (%) | 52 (33.9) | 48 (28.2) | 76 (66.6) | 176 (40.3) |
| | ND, n (%) | 6 (3.9) | 31 (20) | 9 (7.8) | 46 (10.5) |
| | Mean, [interval] | 25.8 (25–89) | 18.9 (13–77) | 35.4 (32–87) | 24.9 (23–89) |
| Survival | Survivors, n (%) | 117 (76.5) | 169 (99.4) | 117 (100) | 403 (92.2) |
| | Deaths, n (%) | 21 (13.7) | 0 (0) | 0 (0) | 21 (4.8) |
| | ND, n (%) | 15 (9.8) | 1 (0.6) | 0 (0) | 16 (3.6) |
| Gender | Male, n (%) | 86 (56.2) | 71 (41.7) | 50 (42.7) | 207 (47.4) |
| | Female, n (%) | 55 (36) | 81 (47.6) | 58 (49.6) | 194 (44.4) |
| | ND, n (%) | 12 (7.8) | 18 (10.7) | 9 (7.7) | 39 (8.9) |
| Locality | Dakar, n (%) | 82 (53.6) | 67 (39.4) | 114 (97.4) | 263 (60.2) |
| | Tambacounda/Kolda, n (%) | 62 (40.4) | 85 (50) | 0 (0) | 147 (33.6) |
| | Saint-Louis/Louga, n (%) | 9 (6) | 18 (10.6) | 0 (0) | 27 (6.2) |

tests. We found a difference in the level of parasitaemia between SM and UM groups, which was statistically significant ($p < 0.01$) with high-level parasitemia in the SM group (mean $\pm$ SD = 27220 $\pm$ 5458.3 P/µL). The haematological parameters of the three groups were compared, as shown in Table 2. Several blood cell indices, such as RBC (red blood cells) count, Hematocrit, MCV, and MCHC, differed significantly between malaria groups. We noted anaemia in the SM group with haemoglobin levels <10 g/dl of red blood cells and platelet counts were significantly lower ($p$-value < 0.0012). In contrast, the leucocytes count was significantly increased ($p$-value = 0.001) in SM compared to UM patients. In addition, we found higher levels of eosinophils, basophils and eosinophils in UM participants than SM participants ($p$-value < 0.01).

### *G6PD* and *HBB* polymorphisms and structure in the Senegalese population

To explore the prevalence of *G6PD* and *HBB* deficiencies in our cohort, the polymorphism of *G6PD* (Chr.X) and *HBB* (Chr.11) genes and their flanking sequences were analysed in 437 Senegalese populations, including 153 SM, 170 UM and 117 CTR participants. Tables 3 and 4 summarise the frequency of each polymorphism in the SM, UM, and CTR groups living in three ecological areas. Polymorphisms were obtained both in forward and reverse sequencing, and allelic frequencies obtained in this study are like the data provided by the NCBI dbSNP database about the African population.

For the *G6PD* gene, PCR amplicon targeted *G6PD*-202 polymorphism amongst the commonly known SNPs in Western Africa and omitted polymorphisms such as *G6PD* +376, *G6PD* +542, *G6PD* +680 and *G6PD* +968. A set of 6 SNP were identified in an

**Table 2  Haematological and Parasitaemia parameters of the study participants.** The mean and SD (Standard Deviation) values of selected haematological parameters and parasite density were determined for three SM, UM and CTR groups. The parasitemia and haematological parameters of the three groups were compared using ANOVA test. Values were statistically significant when compared SM to UM with $p < 0.001$.

| Parameters | Severe Malaria (SM) Mean ± SD | Uncomplicated Malaria (UM) Mean ± SD | Control (CTR) Mean ± SD | P-value (SM/UM) |
|---|---|---|---|---|
| Parasitaemia (P/μL) | 27220 ± 5458.3 | 3993 ± 1327.3 | 0.00 ± 0.0 | <0.001 |
| Leukocytes ($\times 10^3$/μL) | 12.9 ± 1.0 | 8.8 ± 0.5 | 6.1 ± 0.2 | <0.001 |
| Neutrophils (%) | 52.02 ± 3.7 | 60.29 ± 2.5 | 45.77 ± 1.3 | <0.001 |
| Lymphocytes (%) | 22.32 ± 2.4 | 26.21 ± 2.1 | 40.09 ± 1.3 | <0.001 |
| Monocytes (%) | 6.27 ± 0.9 | 7.56 ± 0.9 | 8.65 ± 0.3 | 0.009 |
| Eosinophils (%) | 0.49 ± 0.1 | 2.77 ± 0.48 | 4.33 ± 0.45 | 0.0098 |
| Basophils (%) | 0.44 ± 0.1 | 0.72 ± 0.1 | 1.14 ± 0.08 | <0.001 |
| Red Blood Cells ($\times 10^6$/μL) | 3.71 ± 0.29 | 4.11 ± 0.07 | 4.66 ± 0.06 | 0.0012 |
| Hb[a] (g/dL) | 9.49 ± 0.3 | 12.11 ± 0.2 | 12.96 ± 0.2 | <0.001 |
| Hematocrit (%) | 27.88 ± 0.9 | 36.68 ± 0.7 | 38.44 ± 0.5 | <0.001 |
| MCV (fL) | 79.96 ± 0.64 | 87.70 ± 1.37 | 82.98 ± 0.64 | <0.001 |
| MCHC (pg/cell) | 26.48 ± 0.47 | 29.26 ± 0.47 | 28.04 ± 0.23 | <0.001 |
| Platelets ($\times 10^3$/μL) | 124.1 ± 7.6 | 241.8 ± 5.2 | 286.8 ± 11.2 | <0.001 |

**Notes.**

Hb levels and parasite density distribution are given with the mean and SD values (standard deviation). Hb* values statistically significant when compared SM to UM with $p < 0.001$. Parasite density* $p < 0.001$ when comparing SM to UM.

amplicon of 701pb (Table 3). Among them, 3 SNPs +10707 G > A, +10776 G > C, and +10983 G > C was newly identified genetic variants with low frequencies (MAF <3%). *G6PD* deficiency polymorphisms such as *G6PD*-202 G > A were 0.022, 0.032 and 0.018 in SM, UM and CTR groups, respectively. Another *G6PD* deficiency allele, such as +10588A/G (rs762515), was observed at a high frequency of >3% in the overall population. In specific clinical phenotype populations, the MAF were 0.28, 0.37 and 0.32 in SM, UM, and control groups. There was no evidence of genotypic deviation from HWE except +10588_A > G (rs762515).

Rather than *G6PD*, the *HBB* gene showed more polymorphisms in the Senegalese population with 12 characterised SNPs. These identified SNPs were previously registered in the NCBI database, and rs numbers were already attributed. Then our study does not reveal any novel *HBB* polymorphisms. Among them, 6 SNP (rs7946748, rs7480526, rs10768683, rs35209591, rs334 (HbS) and rs713040) were detected with high frequencies (with MAF > 3%), unlike 6 other SNPs (rs33945777, rs111851677, rs33930165(HbC), rs34598529, rs72561473, rs33944208) with MAF < 3% were observed (Table 4). The MAF of the sickle cell HbS polymorphism was estimated to be 0.026, 0.069 and 0.035, and HbC polymorphism was estimated to be 0, 0.009, 0.029, in SM, UM and CTR groups, respectively.

The structure of *G6PD* and *HBB* genes in the Senegalese population was investigated. The presence of a haploblock structure gene was analysed by measuring the pairwise linkage disequilibrium (LD) in each pair of polymorphisms. Figure 2 shows the values of D' between SNP. For *G6PD* rs762515 A > G, rs1050828 G > A, and for *HBB* rs10768683

Thiam et al. (2022), *PeerJ*, DOI 10.7717/peerj.13487

**Table 3  Frequencies and single nucleotide polymorphism (SNP) of G6PD gene and association analysis with susceptibility to severe malaria (SM).**

| G6PD-SNP | NCBI dbSNP number | Phénotype | MAF | | | | HWE | | | |
|---|---|---|---|---|---|---|---|---|---|---|
| | | | SM | UM | CTR | Overall Population | SM *vs* UM | | UM *vs* CTR | SM *vs* CTR |
| | | | | | | | *P-value* | OR (95% IC) | *P-value* | *P-value* |
| +10588_A >G | rs762515 | | 0.280 | 0.373 | 0.325 | 0.329 | 0.047 | 0.65 (0.43–0.99) | 0.344 | 0.415 |
| +10707_G >A | newSNP | | 0 | 0.004 | 0 | 0.001 | 1 | | 1 | 1 |
| +10776_G >C | newSNP | | 0 | 0.008 | 0 | 0.003 | 0.509 | | 0.517 | 1 |
| +10869_G >A | rs1050828 | G6PD202A | 0.022 | 0.032 | 0.018 | 0.026 | 0.568 | | 0.536 | 1 |
| +10983_G >C | newSNP | | 0.005 | 0.008 | 0.012 | 0.008 | 1 | | 1 | 0.607 |
| +11000_C >T | rs782500951 | | 0.005 | 0.004 | 0 | 0.003 | 1 | | 1 | 1 |

**Notes.**

The *p* values for statistical tests were performed using the linear regression model analysis for each polymorphisms. Association analysis G6PD for polymorphisms (Table 3) and HBB polymorphisms (Table 4) were performed separately, and corrections to each other were applied to reflect the real effect of each. Analysis has been carried out by comparing SM *vs* UM, UM *vs* CTR, and SM *vs* CTR. Borderline ($0.05 \leq p \leq 0.1$) and significant ($0- \leq p \leq 0.05$) *p* values are in bold. The OR (odds ratio) and CI (Confidence intervals) were shown when *p* values were significant. The phenotype attributed by from each polymorphism was mentioned (from NCBI) as: Erythrocytosis 6 familial (E6 familial), Beta-thalassemia (β-thalassemia), Fetal hemoglobin quantitative trait loci 1 (Fetal HQTL-1), Heinz body anaemia (H-B-A), METHEMOGLOBINEMIA (MTH). The SNP labelled as "benign" were 'non-affected SNP', then non associated with many diseases such as Thalassemia, HbSS disease, and/or any other pathology with a notorious phenotype.

SM, Severe Malaria; UM, Uncomplicated Malaria; CTR, Control group; MAF, Minor Allele Frequency; HWE, Hardy–Weinberg.

**Table 4 Frequencies of single nucleotide polymorphism (SNP) of HBB gene and association analysis with susceptibility to severe malaria (SM).**

| HBB-SNP | NCBI dbSNP number | Phénotype (NCBI) | MAF | | | | HWE | | | | |
|---|---|---|---|---|---|---|---|---|---|---|---|
| | | | | | | | *SM vs UM* | | *UM vs CTR* | | *SM vs CTR* |
| | | | SM | UM | CTR | Overall population | *P-value* | OR (95% CI) | *P-value* | OR (95% CI) | *P-value* |
| +526_C >T | rs7946748 | β-thalassemia | 0.06 | 0.03 | 0.06 | 0.05 | 0.06 | | *0.048* | *0.41 (0.17–0.97)* | 1 |
| +519_T >G | rs7480526 | Benign | 0.34 | 0.40 | 0.33 | 0.36 | 0.14 | | 0.14 | | 1 |
| +461_C >G | rs10768683 | Benign | 0.09 | 0.06 | 0.09 | 0.08 | 0.26 | | 0.14 | | 0.75 |
| +446_G >A | rs33945777 | E6 familial, Fetal HQTL-1, Hb SS disease, H-B-A, MTH β type, α- β-thalassemia. | 0.008 | 0 | 0 | 0.002 | 0.19 | | 1 | | 0.50 |
| +436_G >T | rs35209591 | E6 familial | 0.39 | 0.41 | 0.42 | 0.41 | 0.61 | | 0.93 | | 0.56 |
| +200_T >C | rs111851677 | β-thalassemia | 0.02 | 0.03 | 0.02 | 0.02 | 0.79 | | 1 | | 1 |
| +20_A >T | rs334 | HbS | 0.03 | 0.07 | 0.03 | 0.05 | *0.03* | *0.38 (0.16-0.91)* | 0.26 | | 0.45 |
| +19_G >A | rs33930165 | HbC | 0 | 0.01 | 0.03 | 0.01 | 0.26 | | 0.17 | | *0.008* |
| +9_C >T | rs713040 | Hemoglobin OKAYAMA | 0.11 | 0.07 | 0.12 | 0.10 | 0.10 | | *0.03* | *0.51 (0.27-0.93)* | 0.66 |
| −79_A >G | rs34598529 | E6 familial, Fetal HQTL-1, Hb SS disease, H-B-A, MTH β type, α- β-thalassemia. | 0.004 | 0 | 0 | 0.001 | 0.44 | | 1 | | 1 |
| −133_G >A | rs72561473 | β-thalassemia | 0.012 | 0.019 | 0.010 | 0.014 | 0.737 | | 0.486 | | 1 |
| −138_C >T | rs33944208 | β-thalassemia, Hemoglobinopathy | 0 | 0 | 0.005 | 0.002 | 1 | | 0.159 | | 0.206 |

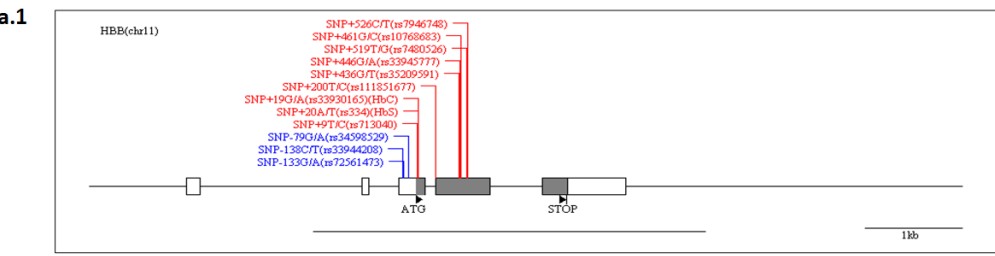

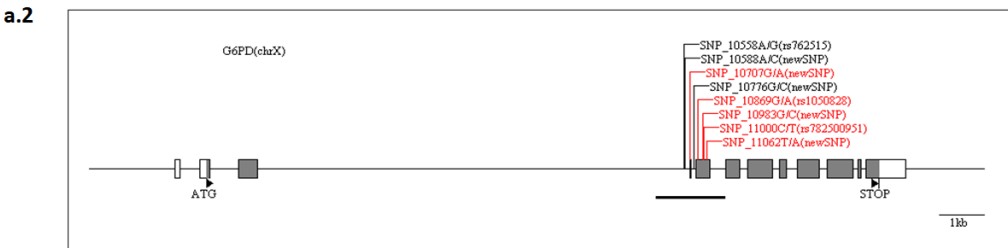

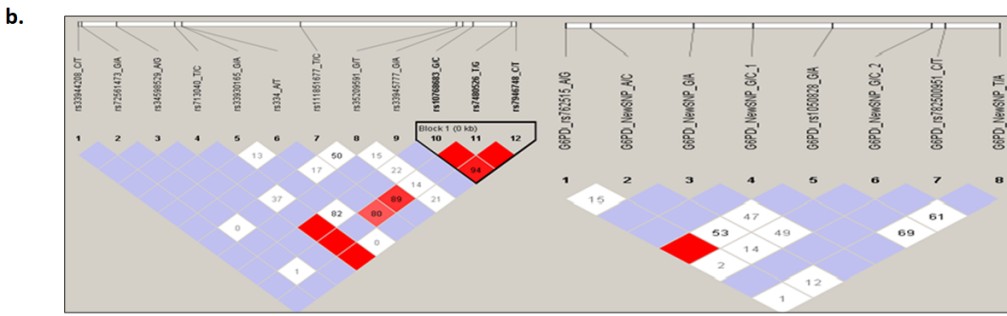

**Figure 2** **The *HBB* and G6PD polymorphisms located on chromosome chr11 and chr X, respectively.** The Haploview software provides the linkage disequilibrium map of HBB and G6PD genes. (A) The coding and UTR regions are indicated by black and white rectangles, respectively, for each loci. SNP positions are numbered per the initiation codon ATG considered as +1 (indicated by a black triangle). The genomic sequence used for alignment is GenBank sequence; reference genes were NG_009015.2 and NG_059281 for HBB and G6PD, respectively. The different colors are: label blue: SNP 'promoter' region, label 'Black', SNP'intronic' region and label 'Red', 'exonic' region. An underlined black line shows the boundaries of PCR amplicon in each gene. (B) The linkage disequilibrium map of LD plots within regions of 800 and 700 bases for HBB and G6PD genes, respectively. The LD plots show pairwise D′ values given in diamonds for each statistical comparison between the SNPs. The different shades of color represent D′ values (between 0 and 1). In white and red 'diamond', the values of D′ are indicated, an empty red diamond indicates that D′ > 0.97.

G > C, rs7480526 T > G, rs7946748 C > T polymorphisms, the values D' > 0.97, showing a high linkage disequilibrium, and defining a block structure in *HBB* genes. A genetic segment appeared to be part of a haploblock structure in the *HBB* loci, composed of 3 'successive' SNPs, for which the confidence interval D′ was 0.9–1 (rs10768683 G > C, rs7480526 T > G, and rs7946748 C > T).

### *G6PD*, *HBB* polymorphisms and clinical malaria

To test whether *G6PD* and *HBB* polymorphisms were associated with malaria protective effect in our overall population, a statistical association analysis was performed using

logistic regression by comparing the three phenotypes groups, SM, UM and underage, sex as covariates. For *G6PD*, the SNP +10588 A > G (rs762515) yielded a significant association with protection from severe malaria in Senegalese populations. For SM *vs* UM, the SNP +10588 A > G (rs762515) showed a borderline *p*-value = 0.047 (OR 0.65, 95% CI [0.43–0.99]). For *G6PD*-202 G > A, no association with protection against severe malaria was found in our overall population (Table 3).

An analysis of HbS polymorphism showed a significant association with protection against severe malaria. For SM *vs.* UM comparison, the sickle cell trait HbS polymorphism (*HBB* +20 A > T, rs334) yielded a significant *p-value* = 0.033 (OR 0.38, 95% CI [0.16–0.91]) (Table 4). Surprisingly, the rs33930165 (HbC) polymorphism is not a protective variant against severe forms in our population. There is no significant difference between SM *vs* UM, with a *p*-value = 0.26 (Table 4).

### *G6PD*, *HBB* polymorphisms and ecological malaria areas

We then performed comparisons of frequencies among the known malaria protective SNPs in 3 ecological areas and tried to distinguish relations with endemicity. The SNPs HbS (rs334), HbC (rs33930165) and *G6PD* +10588A > G (rs762515) were associated with protection against severe malaria in the three regions by comparison of SM *vs* UM with higher frequency in UM group (Fig. 3). However, *HBB* +9 C > T (rs713040) has variable protective effects depending on endemicity. In St-Louis, a low endemicity area, the SNP rs713040 and rs7946748 were associated with protecting the severe form, unlike Dakar and Tambacounda/Kolda/, which are more endemic areas. For *G6PD*-202 G > A, we found differences per the locality. The polymorphism was associated with protection against SM in the Dakar Capital centre but not in the other regions (St Louis-Louga and Tambacounda-Kolda). Indeed, the SNP is associated with the protection against SM form in Dakar but not in the other regions (Fig. 3).

### *G6PD*, *HBB* polymorphisms and biological parameters

Analyses were conducted to test whether *G6PD* and *HBB* polymorphisms were associated with biological parameters of severity, performing comparisons of *SM vs UM*. Then correlation of *HBB* polymorphisms *HBB* +226 C > T (rs7946748), HbS (rs334), HbC (rs33930165), *HBB* +9 C > T (rs713040), and *G6PD*-202 G > A with age, parasitaemia, and biological parameters (hemoglobin levels, blood platelets, lymphocytes, monocytes, basophils, neutrophils and eosinophils) and survival/death outcome, were performed using the Mann–Whitney test.

Our results showed a significant association between age and *HBB* +200_T > C (rs111851677), a variant responsible for β-thalassemia.

Then results showed significant association of rs7946748 with basophils levels *p-value = 0.01*5 (beta = −0.5458, 95% CI [−0.85 to −0.11]) and lymphocytes *p-value = 0.04* (beta = −8.59, 95% CI [−16.73 to −0.44]) (Fig. 4); the rs10768683 and rs713040 were associated to basophils with *p-value = 0.023* (beta =−0.38, 95% CI [−0.7 to −0.06]) and *p-value = 0.020* (beta = −0.37, 95% CI [−0.6 to −0.06]) respectively. HbS (rs334) triggered a significant association with eosinophils *p*-value = 0.024 (beta = 3.532, 95%
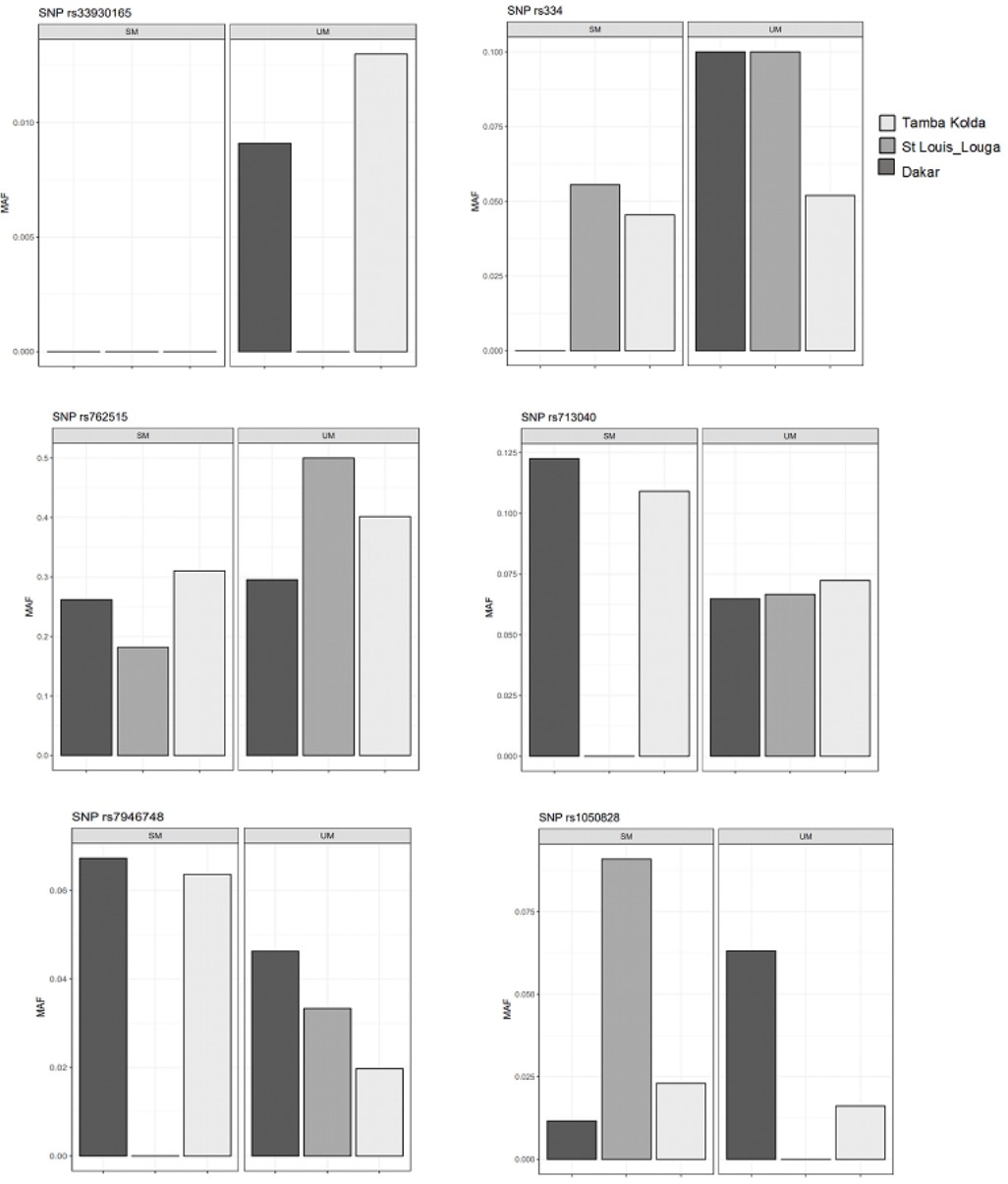

**Figure 3  Allelic frequencies of G6PD and HBB in different study areas.** Light grey, black grey and black areas indicate allelic frequencies from Dakar, Saint-Louis_Louga and Tamba_Kolda, respectively.

CI [0.51–6.54]). Platelets were associated with *G6PD*-SNP +10707 G > A, *G6PD*-SNP +10983 G > C and *HBB*-rs72561473, with $p = 0.009$, 0.048, 0.006 respectively. Finally, *G6PD*- SNP +11000_C > T (rs782500951) was associated with parasitaemia with a *p* value $= 0.016$ (Fig. 4).

| Biological parameter | SNP | chr | BP | A1 | BETA | SE | PV |
|---|---|---|---|---|---|---|---|
| Basophils | rs7946748 | 11 | 5226496 | T | -0.5458 | 0.219 | 0.01523 |
| | rs10768683 | 11 | 5226561 | G | -0.3793 | 0.1637 | 0.02359 |
| | rs713040 | 11 | 5227013 | T | -0.3776 | 0.1586 | 0.02018 |
| Lymphocytes | rs7948748 | 11 | 5226496 | T | -8.59 | 4.154 | 0.04193 |
| Esosinophils | rs7480526 | 11 | 5226503 | G | 1.32 | 0.4156 | 0.002253 |
| | rs334 | 11 | 5227002 | T | 3.532 | 1.539 | 0.02485 |
| Platelets | rs72561473 | 11 | 5227154 | A | 199.9 | 76.22 | 0.009376 |
| | SNP_10707_GA | X | 10707 | A | 287 | 144.4 | 0.0483 |
| | SNP_10983_GC | X | 10983 | C | 175.4 | 64.26 | 0.006901 |
| Parasitaemia | rs782500951 | X | 11000 | T | 0.00001 | 0.0004309 | 0.01681 |
| Age | rs111851677 | 11 | 5226822 | C | 11.94 | 5.548 | 0.03244 |

**Figure 4   Forest plot of associations between HBB and G6PD polymorphisms and biological and cellular parameters.** Multivariate linear regression analysis with biological parameters. The blue squares represent βeta values. The horizontal lines around squares represent the 95% Confidence Intervals. SNP, single nucleotide polymorphism; MAF, minor allele frequency; beta, regression coefficient corresponding to the minor allele of each SNP; SE, Standard Error; BP, genomic location; PV, *p*-value; CHR, chromosome code; A1, Allele 1 (usually minor).

## DISCUSSION

In our present study, we characterised the polymorphisms of *G6PD* and *HBB* genes in the Senegalese population, two-loci previously associated with protection against severe malaria. A genetic analysis was performed to identify polymorphisms associated with clinical malaria and their prevalence, using three specific phenotypes of severity, including SM, UM and CTR groups living, from North to South, in three ecological malaria transmission areas.

We identified six polymorphisms on *G6PD* genes and found that the minor allele frequency of *G6PD*-202 G > A was 0.022, 0.032, 0.018 in SM, UM, and CTR groups, respectively. The frequency of these polymorphisms was 0.026 in the overall population. In The Gambia, a country neighbouring Senegal, it was reported that the minor allele frequency of this polymorphism in the CTR sample was 0.03, considerably lower than reports from other east African countries such as Kenya (0.18) and Malawi (0.19) (*Clarke et al., 2017*).

Our results confirm the weak prevalence of allele deficiency from *G6PD*-202 G > A in the Senegambian area compared to Kenya and Malawi populations. These results could ensure that *G6PD* polymorphisms prevalence is highly population-specific, and thus, attention should be paid to the study population from which future estimates may be drawn (*Galatas*

*et al., 2017*). In our Senegalese population cohort, we found that *G6PD*-202 G > A polymorphism was not associated with protection against severe malaria at the global level. However, differences were founded in the different ecological localities. The polymorphism was associated with protection against SM in Dakar Capital centre but not in the other regions. Our results could contradict published data. Indeed the A-allele of *G6PD*-202 G > A polymorphism has been associated with protection against severe malaria in African populations, such as in Mali, the Gambia and Uganda (*Guindo et al., 2007*; *Ruwende et al., 1995*). However, this protective effect seems to depend on other parameters such as sex and locality. As reported by *Maiga et al. (2014)* the studies have shown the protective effect against severe malaria has not been consistent across extensive studies observed in females (*Uyoga et al., 2015*; *Manjurano et al., 2015*), in males (*Guindo et al., 2007*), in both (*Ruwende et al., 1995*), or no protection (*Clark et al., 2009*; *Toure et al., 2012*). This disparity between the results of the studies could be explained in part by variation in phenotype definition, choice of controls (village surveys *vs* hospital-based studies), age or immune status of participants, and study designs (case-control *vs* cohort) (*Clark et al., 2009*; *Maiga et al., 2014*). *G6PD* deficiency can potentially protect against uncomplicated malaria in African countries, but not severe malaria. Interestingly, this protection was mainly heterozygous, being x-linked and thus related to gender (*Guindo et al., 2007*; *Mbanefo et al., 2017*). No such protection was evident from the mosaic state of *G6PD* deficiency in heterozygous females. A previous study confirmed highly significant protection against severe malaria in hemizygous males but not in heterozygous females (*Guindo et al., 2007*). A case-control study showed a significant association between *G6PD* A- and risk of severe malaria, with protection against cerebral malaria but increased risk of severe anemia (*Clarke et al., 2017*). However, compared to HbAS or α–thalassemia, associations between *G6PD* A- deficiency and risk of severe malaria have been less straightforward, with studies yielding inconsistent results (*Manjurano et al., 2015*; *Mbanefo et al., 2017*). In fact, it has been shown that the *G6PD*-202 polymorphism may not be a good marker of A- deficiency, and/or other polymorphisms are required to confirm the protective effect (*Clark et al., 2009*). Our study found that the *G6PD* SNP +10588_A > G (rs762515) yielded a significant association with protection from severe malaria in the Senegalese population.

In the *HBB* genes located on chromosome 11, we identified 12 polymorphisms and characterised their frequencies. The SNP HbS resulted in substituting a glutamate residue with a valine or lysine residue in the ß-globin chain at position 6. The Minor Allele Frequency of HbS and HbC polymorphisms was estimated to be 0.05 and 0.01, respectively, in the overall population. Several studies have shown a low frequency of the HbS allele in the control group (∼3.8%). The frequency is in keeping with other West African countries (Burkina Faso 5.2%, Cameroon 6.5%, The Gambia 7.6%, and Ghana 6.5%) and east African populations (Kenya 6.4%, Malawi 2.7%, Tanzania 7.8%) (https://malariaatlas.org/explorer/#/). A higher frequency of the HbC allele has been observed in other West African populations (*Agarwal et al., 2000*; *Toure et al., 2012*).

Our analysis confirmed the known protective effects of HbS. We found that HbS polymorphism showed a significant association with protection against severe malaria in the three ecological regions. For SM *vs.* UM, the sickle cell trait HbS polymorphism (*HBB*

+20_A > T, rs334) yielded a significant $p$-value = 0.033. The observed frequencies for HbS (rs334) are 0.026, 0.069 and 0.035 for SM, UM, and CTR groups, respectively (Table 4). These data fit with studies and dogma on the protective effect of HbS polymorphism, corresponding to a lower frequency in 'severe malaria form' and an increase in the 'uncomplicated' malaria group and 'healthy' control subject, representing the general population (Table 4) (*Lopera-Mesa et al., 2015*; *Kreuels et al., 2010*; *Amodu et al., 2012*). In fact, the differences were not significant when comparing UM*vs* CTR, and SM*vs* CTR, probably to the fact of small numbers of controls and then, consequently, a lack of power.

Unexpectedly, the protective effect of the HbC variant has not been observed in our analysis. The HbC polymorphism (*HBB* +19_G > A, rs33930165) not showed significant; we found a $p$-value = 0.26 when comparing SM *vs*. UM. Previous studies showed that HbAS had been associated with 50% and 80% reduced risks of developing uncomplicated and severe malaria (*Williams, 2005b*); HbAC and HbCC have been associated with 30% and 93% reduced risks of developing mild and severe malaria, respectively (*Modiano et al., 2001*). A study conducted in Mali, a neighbouring country of Senegal, indicated that HbC is associated with protection against the SM form of *P falciparum* malaria in the Dogon of Bandiagara population (*Clarke et al., 2017*). In this ethnic group, the prevalence of HbC was significantly lower among severe malaria cases than among cases of uncomplicated malaria. The protective effect indicated an 80% reduction in the risk of severe malaria. The data also suggest that the protective effect associated with HbC may be greater than HbS in this population (*Agarwal et al., 2000*).

It was suggested a "balanced polymorphism" where the HbS homozygote disadvantage is repaid through the resistance of the heterozygote HbAS in regions where malaria is endemic (*Lederberg, 1999*).

Our results suggest differences in the protective effect of hemoglobin among the three ecological areas. The possibility that protective effects associated with different hemoglobin mutations vary among diverse human populations. This is consistent with the report's findings. In fact, in West Africa, the Fulani ethnic group has decreased susceptibility to malaria compared to Dogon populations (*Bereczky et al., 2006*), and the Fulani also have a reduced prevalence of sickle hemoglobin (*Nasr et al., 2008*), α-thalassemia (*Modiano et al., 2001*), and *G6PD* A- (*Modiano et al., 2001*; *Maiga et al., 2014*) compared to other groups.

We have also observed a positive association between *HBB* +200_T > C (rs111851677) variant and age. Moreover, this variant does not have a protective effect against severe malaria in our study. Age is an essential factor in the pathophysiology of malaria, and some reports from sub-Saharan Africa suggest that *P. falciparum* infections are more common among children aged 5–15 years than among younger children and adults (*Cohee et al., 2020*; *Were et al., 2018*; *Toure et al., 2016*). However, children are under-represented in our cohort because clinical forms were included without any preconceived notion of age categories. There are more adults because our inclusions are not oriented toward pediatric services; to fill this gap, this will be a point to consider in our future inclusions.

The results showed significant associations between several polymorphisms on *HBB* and *G6PD* genes, parasitemia, and white blood cells (basophils, eosinophils, and lymphocytes).

*P. falciparum* infection is associated with a profound T-cell activation with the Th1-derived cytokine interferon-$\gamma$ as a major pathogenic factor (*Hunt & Grau, 2003*; *Belnoue et al., 2008*). Eosinophils and basophils seem to play important roles in malaria pathogenesis. Increases in plasma and tissue histamine derived from basophils have been associated with disease severity in human *P. falciparum* infections and several animal malaria (*MacDonald et al., 2001*). In addition, elevated plasma levels of IgE, which binds to basophils, have been associated with the severity of *P. falciparum* infection (*Pelleau et al., 2012*; *Porcherie et al., 2011*). Furthermore, increased eosinophil counts have been associated with recovery from infection (*Camacho et al., 1999*; *Pollenus et al., 2020*). One of our recent studies showed that Ribonuclease 3 (RNASE 3), also known as eosinophil cationic protein (ECP), is one of the factors suggested as having a role in malaria severity. The polymorphisms: +371G/C (rs2073342), +499G/C (rs2233860) and +577A/T (rs8019343) defined a haplotype risk (G-G-T) for malaria severity (*Diop et al., 2018*). Despite these observations, little is known about the relationship between these cell types and disease. Our results showed that rs334 (HbS) is a variant associated with eosinophils (beta =3.532, 95% CI [0.51–6.54], *p*-value = 0.024) (Fig. 4). It would be interesting to study this association in-depth in future studies and compare eosinophils' levels in patients with HBSS and HbAS variants.

The early and vigorous development of malaria immunity has been associated with a protective effect of sickle cell traits and thalassemia. Although, ethnic variations in the immunological response to *P falciparum* infection and rates of the clinical episodes of malaria are described in clinical studies in West Africa (*Modiano et al., 2001*; *Agarwal et al., 2000*).

In conclusion, our study gives an overview of the importance of sickle cell polymorphism on severe malaria susceptibility in the Senegalese population.

## ACKNOWLEDGEMENTS

The authors are grateful to all the patients and the medical staff who have generously collaborated in the Malaria Genomic Project Clayton Dedonder 2014 (Institut Pasteur de Dakar). The authors thank the staff of the Centre National de Recherche en Génétique humaine (CNRGH) for assistance in performing the genotyping. The CNRGH-CEA performed all sequencing of the target genes free of charge.

### Funding
This study was supported by grants from Dedonder Clayton 2014, Institut Pasteur Network. The funders had no role in study design, data collection and analysis, decision to publish, or preparation of the manuscript.

### Grant Disclosures
The following grant information was disclosed by the authors:
Dedonder Clayton 2014, Institut Pasteur Network.

## Competing Interests

The authors declare there are no competing interests.

## Author Contributions

- Fatou Thiam conceived and designed the experiments, performed the experiments, analyzed the data, prepared figures and/or tables, authored or reviewed drafts of the article, and approved the final draft.
- Gora Diop conceived and designed the experiments, performed the experiments, analyzed the data, prepared figures and/or tables, authored or reviewed drafts of the article, and approved the final draft.
- Cedric Coulonges analyzed the data, authored or reviewed drafts of the article, and approved the final draft.
- Céline Derbois analyzed the data, authored or reviewed drafts of the article, and approved the final draft.
- Babacar Mbengue conceived and designed the experiments, performed the experiments, authored or reviewed drafts of the article, and approved the final draft.
- Alassane Thiam conceived and designed the experiments, performed the experiments, authored or reviewed drafts of the article, and approved the final draft.
- Cheikh Momar Nguer conceived and designed the experiments, performed the experiments, prepared figures and/or tables, authored or reviewed drafts of the article, and approved the final draft.
- Jean Francois Zagury analyzed the data, authored or reviewed drafts of the article, and approved the final draft.
- Jean-Francois Deleuze analyzed the data, authored or reviewed drafts of the article, and approved the final draft.
- Alioune Dieye conceived and designed the experiments, performed the experiments, authored or reviewed drafts of the article, and approved the final draft.

## Human Ethics

The following information was supplied relating to ethical approvals (*i.e.*, approving body and any reference numbers):

UCAD Ethic committee (University Cheikh Anta DIOP, Senegal) (#Protocole 0344/2018/CER-UCAD#).

## DNA Deposition

The following information was supplied regarding the deposition of DNA sequences:

The SNPs are available at GenBank: rs762515, rs1050828, rs782500951, rs7946748, rs7480526, rs10768683, rs33945777, rs35209591, rs111851677, rs334, rs33930165, rs713040, rs34598529, rs72561473, rs33944208.

## Data Availability

The raw data and sequences are available in the Supplementary Files.

## Supplemental Information

Supplemental information for this article can be found online at http://dx.doi.org/10.7717/peerj.13487#supplemental-information.

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
