# Peer review of "G6PD and HBB polymorphisms in the Senegalese population: prevalence, correlation with clinical malaria"

_PeerJ, doi:10.7717/peerj.13487_

## Round 0.1 · original submission · Major Revisions

Your manuscript was considered interesting and valuable by the reviewers, however, there are a number of issues they raised that need to be addressed. Two of the reviewers had concerns about your sample size, the rationale behind establishing your sample size, and why your sample mostly included adults when most of the malaria burden is in children younger than 15. They were also concerned that age was not taken into account in your study, particularly when comparing areas of different malaria endemicity. The reviewers would also like you to state the number of controls in the abstract, include more details on subject recruitment, and explain how 323 samples are globally representative of the Senegalese population as you state. The reviewers also wanted additional clarification for your figures and tables, as well as for your methods and statistical analysis. For example, they would like to know the criteria you used to define uncomplicated vs severe malaria and whether HBC has a protective effect or not. Lastly, one of the reviewers requested that you improve the flow of the background section, and several reviewers suggested that you improve the English language in the manuscript, as well as check for spelling and grammatical errors.

Please, submit a detailed rebuttal that shows where and how you have taken all comments and suggestions into consideration. If you do not agree with some of the reviewers’ comments or suggestions, please explain why. Your rebuttal will be critical in making a final decision on your manuscript. Please, note also that your revised version may enter a new round of review by the same or by different reviewers. Therefore, I cannot guarantee that your manuscript will eventually be accepted.

Reviewer 1 ·

Basic reporting

no comment

Experimental design

In the manuscript, the authors were aimed to investigated the prevalence of HBB and G6PD genes polymorphisms in Senegalese populations to evaluate the risk for severe Plasmodium falciparum malaria and its specific phenotypes. The study was well designed and the results were sufficient to comment about the affect of the studied gene polymorphisms on Malaria clinical phenotype.

Validity of the findings

the results were sufficient to comment about the affect of the studied gene polymorphisms on Malaria clinical phenotype as well as the authors had selected suitable statistical analysis.

Additional comments

no comment

Reviewer 2 ·

Basic reporting

1. The manuscript lay out conforms with that required for PeerJ. Whilst the English is overall very good, there are innumerable spelling and small grammatical errors throughout. In many places we have random capital letters mid sentence. There are also some strange sentences dotted throughout. For example Line 216 “The materiality threshold was set at p=005.” Could do with another very very thorough read through.
2. Human gene abbreviations should be in italics. This has been done occasionally but not throughout.
3. In the results section on blood indices, it is written in Table and text PNB and PNE. Can these be spelled out.

Experimental design

1. The study sets out two objectives, the first of which is “to investigate the prevalence of key malaria-protective 138 polymorphisms in G6PD and HBB genes in the global Senegalese population”.
I’m not convinced that the samples from 323 individuals can really be considered globally representative of the Senegalese population, especially, as well described in the paper, the large number of ethnic groups.
2. And in the methods it states that “114 CTR, 170 UM and 153 SM subjects were included” which sums to over 400. Maybe best to state the number of controls in the abstract as well (ie 437 individuals not just 323)
3. Line 219-224. This is really not very clear and surprised to see a result given in the methods section. It’s hard to understand what analysis was done but it doesn’t look correct.
“The participant’s average percentage parasitaemia is calculated by taking the parasitaemia reported by all included subjects and calculating the average. From those results, the standard deviation is determined, and the results>3 standard deviations from the average are omitted and a new mean and standard deviation calculated. This new average is referred to as the mean parasitamia. We found a difference of level of parasitaemia between SM and UM groups, which were statistically significant (P < 0.01).”
4. Line 252 “The PCR products were checked by 1.5% agarose gel electrophoresis wholly,….”
What does this mean?
5. Line 303 2.2. Haemotological and parasitaemia parameters of the study population.
In this small paragraph there’s no mention of parasitemia. Maybe what was written in the methods should have been placed here.
It would be a good idea for the reader to define precisely what criteria are required for a classification as Uncomplicated or Severe malaria.

Validity of the findings

1. Age is a hugely important factor to be taken into account for the outcome of malaria infection. I can’t see where this has been taken into account. This is also especially important when comparing areas of differing endemicity.
2. Line 359 onwards. I’m a little concerned about the stat results as we have a significant P value whereas the 95% confidence intervals go over 1. I don’t see how this can be significant. Please see general comment below.
The HbC polymorphism (HBB +19 G>A, rs33930165) showed a significant p value when comparing SM vs. CTR (p value = 0.0084, OR 1.67; 95% IC 0.99–3.42) (Table 4). The comparative analysis UM vs. CTR of the other Hb polymorphisms showed significant associations with rs7946748 C>T and rs713040 C>T, respectively (p value = 0.033, OR 2.67 95% IC: 2.01–3.50) (p-value = 0.008, OR 1.67; 95% IC 363 0.99 - 3.42) (Table 4)
3. There are lots of blood cell parameters that are associated with various SNPs but I can’t see what analyses was performed and worse still these findings are not at all discussed in the discussion.
4. Table 2 In P value column it is written SM/MM. What is MM?

Additional comments

I am a little bewildered about the HBC result. In the abstract it states that there was no protective effect, in the results and discussion it states there was despite the Confidence intervals overlapping with 1 (see above).
As noted above the associations of snps with blood cell parameters could be interesting but we need to know more about these analyses and the co-variates used.

Reviewer 3 ·

Basic reporting

Some editing required across the document. Major editing needed as the text is not in proper English. The raw data has been shared and relevant references have been used.
Line 75-76 Malaria is caused by Plasmodium species infection and affect hundred of millions of people per year
Line 77-78 Malaria disease remain a major cause of death and is still the 4th leading cause of death for infectious diseases worldwide.
Line 81-82 And the comprehension of these mechanisms are necessary for a global malaria eradication
Line 93-94 In endemic areas, the protective effect of HBB variants (HbS and HbC) on P. falciparum malaria clinical phenotypes were reported.
Line 120 and 121. G6pd and G6pD
Line 217 The individual % parasitaemia was mesure as
Line 160 Ethnics distribution
Line 321 SNPs in Western Africa, and lefted out polymorphisms such as
Line 400 In The Gambia, a neighbor country of
Use of comma instead of decimals in numerals is not standard practice.
All abbreviations should be described. What is PNB (line 384) and PNE (line 386)
Line 266 (Ref 28-29 de Zuo) the references need to be quoted appropriately.

Experimental design

Sample size: how was the number of participants arrived at? Most studies have been done in children who are under-represented in this study, what was the rationale for the inclusion of adults?

153 SM and 170 UM cases from 2003-2015 across the regions these are very low numbers is this the actual number of cases or was there selection bias. Where the participants actively recruited across the years then followed up at admission or how exactly is this a cohort? Or where the available stored samples retrieved for the study.

Potential confounders (sex and ethnicity) are listed in the discussion, were any of them adjusted for in the analysis for the ORs?
The significance for SM vs UM rs762515 is borderline would this still hold when analyzed by sex? The severe and uncomplicated malaria cases need to be clearly defined. Is there a parasitaemia threshold?

Is the Plasmodium-positive QBC test quantitative or qualitative? What are the product details?
What is the age of the study participants?
Where were the controls recruited? if at the hospital what was the cause of admission?
Why wasn’t parasitaemia expressed as parasites/uL instead of %? Yet on Table 2 it is presented as parasites/uL
Lines 213-224 are describing statistical analysis hence should move to section 1.5
The tests should be named properly Student's test and Yates corrected X2 test
G6PD has previously been described to be X-Linked what is the relevance of analyzing the sex-specific effects underlying X chromosome loci?
Ethics section should be after section 1.5

Validity of the findings

Mean age of the participants is 25.8 years. The burden of malaria lies in children yet those <15 years of age are only 33% of the participants. Was the study done in the correct age group? What is the burden of malaria in those over the age of 15 in Senegal?

Line 300 “13.7% of mortality was observed.” What % of this mortality occurred in children? What % of the severe malaria cases died?

Table 1 - why were all the controls recruited from Dakar yet 50 %of the cases come from the other regions? The controls are also underrepresented in age <5years category.

Table 2 - what proportion of the participants had severe anemia? What proportion of the participants had bacteraemia? Parasitemia should be presented as geometric mean. Lymphocyte and neutrophil should be presented as actual counts. It’s not clear what the value of monocyte, eosinophil and basophil count is on this table.

“Global population” used in tables 3 and 4 should be replaced with “overall” is the mean that the MAF of the overall population sampled. Same change should be effected in the discussion as the word global is misleading.

Section 2.5 – the investigators are reporting results of SNPs showing protection against malaria yet the data has not been included in the document.

Lines 385-389 – what is the direction of the reported associations?

What is the role of rs7946748? Has it been described before?

Line 407- 411 “In our Senegalese population cohort, we found that G6PD-202 G>A polymorphism was not associated with protection against severe malaria at global level. However, differences were founded in the different ecological localities. The polymorphism was associated with protection against SM in Dakar Capital center, but not in the other regions (St Louis-Louga and Tambacounda-Kolda).” This text is not supported by the data presented by the authors.

Additional comments

The number of controls is missing in the abstract.

Line 79 asymptomatic malaria should be initialed as AM not UM which is normally used for uncomplicated malaria. Mild malaria should be replaced with Uncomplicated malaria UM as is the case in the rest of the paper. Line 399 states MM.

The flow of the background needs revision. Lines 75-92 should be followed by the description of the G6PD and HBB genes before describing the association with malaria endemicity. What is the prevalence of HbC in west Africa?

Consider replacing the word subjects with participants.

Line 41-43 “We investigated the prevalence of HBB (chr11) and G6PD (chrX) deficiencies polymorphisms, among Senegalese populations, the risk for severe Plasmodium falciparum malaria and its specific phenotypes, including severe and mild malaria.”
Mild malaria is a phenotype of malaria not severe malaria.

Line 140-141 We characterized polymorphisms on HBB and G6PD genes in residents from three ecological zones where the malaria endemicity was different.
Did the different ecological zones also have different ethnicity?

Reviewer 4 ·

Basic reporting

The research article entitle "G6PD and HBB polymorphisms in the Senegalese population: prevalence and correlation with clinical malaria" by Fatou Thiam et al. The manuscript is written very well and organized. However, it needs minor correction before final publication. The idea of the study of G6PD and HBB polymorphisms in the Senegalese population is interesting and will enhance knowledge for the development of malaria drugs. I would request authors to please correct sentences and grammar so that the non-science people could easily understand your work and its importance.

The 2nd reference is a very old study. Please replace with some new article and finding and also look other references.

Introduction is very comprehensive and representing sufficient for this study. Covered all manuscript part and follow but please check references and grammar before final submission.

Additional, Authors advised to check the figures number, visibility, figures legend, Table no, and legends carefully before final submission.

Figure 1: Mark the number of samples collected from different. What is represented (0,30,60,120, 180, and 240) in this figure, and how is it important? What means outside (border number) in the figures? Is it relevant with endemic or sample collection? Would you please clarify it?

Figure 2: For better visualization and representation, I suggest plotting a lollipop plot of HBB and G6PD with all exonic and intronic regions. Please also mark all SNPs in 3D protein structure.

Table: Write the full name of MAF, HWE, SM, UM, and CTR below. It will help the reader to understand easily.

Experimental design

The overall finding of this project is a novelty in the context of G6PD and HBB polymorphisms in the Senegalese population, and this work should be publishable after minor revision.


The research questions are well defined and relevant to the scientific community, especially malaria research. Authors used a adequate number of samples to address the questions.

Validity of the findings

"No comment"

---

## Round 0.2 · Minor Revisions

Thank you for thoroughly addressing the reviewers’ comments. The only comment that the reviewers had on your revised manuscript was with regard to improving the English language, and syntax in your manuscript. The reviewer kindly provided detailed comments that they suggested you address.

Please, submit a detailed rebuttal which shows where and how you have taken all comments and suggestions into consideration. If you do not agree with some of the reviewers’ comments or suggestions, please explain why. Your rebuttal will be critical in making a final decision on your manuscript. Please, note also that your revised version may enter a new round of review by the same or by different reviewers. Therefore, I cannot guarantee that your manuscript will eventually be accepted.

Reviewer 2 ·

Basic reporting

Thank you for the extensive revisions. It reads much better. It could do with another read through and a spell check as there are a number of misspelled words and some bizarre use of capitals mid-sentence. Otherwise some minor points below.
Line 51 It reads Results: Six frequent SNPs The minor allele frequency (MAF) > 3%) ….
Maybe should read: Six frequent SNPs with a minor allele frequency (MAF) > 3%
Line 138 It reads “malaria versus outcome” Word missing?
Line 163 “Even there is an ethnic distribution from North to South regions” word(s) missing?
Line 164/5 “For commodity” do you mean convenience?
Line 172 “however, health conditions are different with better disease coverage..” what is better disease coverage?
Line 195 “Still, other complications such as hypoglycemia, thrombocytopenia, renal insufficiency, hepatic or even pulmonary oedema may appear alone or in combination.” This sentence seems to come from nowhere. Do you mean these conditions without malari infections?
Line 252-4 “Differences in allelic frequencies among the three groups (SM, UM, CTR) were implemented using the logistic regression analysis method, performed elsewhere [55, 56].”
This is a strange sentence. What is meant by implemented and performed elsewhere.
Line 489-491 “It is logical that there are more adults. Because our inclusions are not oriented towards pediatric services, otherwise this will be a point to be shaved in our future inclusions.” This needs to be reworded and I’m not sure why “shaved” is used.
Line 511 “Our results showed an association between In conclusion,” There seem to be words missing before the “In conclusion…”
Table 2. Maybe include stat test used in the title.
Table 5 needs more explanation in the legend. I presume PV is P-value but the rest are? And what do the blue boxes, lines and arrows refer to?

Experimental design

Previous issues addressed. Dieureudieuf.

Validity of the findings

Previous issues addressed. Dieureudieuf.

Additional comments

Previous issues addressed. Dieureudieuf.

Reviewer 3 ·

Basic reporting

The authors have taken into account issues raised by reviewers.

Experimental design

No comment

Validity of the findings

No comment

Additional comments

The manuscript reads a lot better than the earlier version.

---

## Round 0.3 · accepted · Accept

Thank you for thoroughly addressing the reviewers' comments. As a result your manuscript is greatly improved.

The Academic Editor noted that the authors used the word "locus" to refer to "loci" so please ensure you correct this in the proof stage.

Reviewer 2 ·

Basic reporting

Points addressed and english corrected

Experimental design

Points addressed.

Validity of the findings

Points addressed

Additional comments

No additional comments

Reviewer 4 ·

Basic reporting

The authors have corrected all the raised comments. Thanks for the quick response. Now research article is impressive, and it could be publishable.

Experimental design

No change is required.

Validity of the findings

Not required any modification original article is fine for publication.